# Involvement in Tumorigenesis and Clinical Significance of CXCL1 in Reproductive Cancers: Breast Cancer, Cervical Cancer, Endometrial Cancer, Ovarian Cancer and Prostate Cancer

**DOI:** 10.3390/ijms24087262

**Published:** 2023-04-14

**Authors:** Jan Korbecki, Mateusz Bosiacki, Katarzyna Barczak, Ryta Łagocka, Agnieszka Brodowska, Dariusz Chlubek, Irena Baranowska-Bosiacka

**Affiliations:** 1Department of Biochemistry and Medical Chemistry, Pomeranian Medical University in Szczecin, Powstańców Wlkp. 72, 70-111 Szczecin, Poland; jan.korbecki@onet.eu (J.K.); bosiacki.m@gmail.com (M.B.);; 2Department of Anatomy and Histology, Collegium Medicum, University of Zielona Góra, Zyty 28 Str., 65-046 Zielona Góra, Poland; 3Department of Functional Diagnostics and Physical Medicine, Faculty of Health Sciences Pomeranian Medical University in Szczecin, Żołnierska 54 Str., 71-210 Szczecin, Poland; 4Department of Conservative Dentistry and Endodontics, Pomeranian Medical University, Powstańców Wlkp. 72, 70-111 Szczecin, Poland; katarzyna.barczak@pum.edu.pl (K.B.);; 5Department of Gynecology, Endocrinology and Gynecological Oncology, Pomeranian Medical University in Szczecin, Unii Lubelskiej 1, 71-252 Szczecin, Poland; agnieszka.brodowska@pum.edu.pl

**Keywords:** chemokine, CXCL1, breast cancer, cervical cancer, endometrial cancer, ovarian cancer, prostate cancer

## Abstract

C-X-C motif chemokine ligand 1 (CXCL1) is a member of the CXC chemokine subfamily and a ligand for CXCR2. Its main function in the immune system is the chemoattraction of neutrophils. However, there is a lack of comprehensive reviews summarizing the significance of CXCL1 in cancer processes. To fill this gap, this work describes the clinical significance and participation of CXCL1 in cancer processes in the most important reproductive cancers: breast cancer, cervical cancer, endometrial cancer, ovarian cancer, and prostate cancer. The focus is on both clinical aspects and the significance of CXCL1 in molecular cancer processes. We describe the association of CXCL1 with clinical features of tumors, including prognosis, ER, PR and HER2 status, and TNM stage. We present the molecular contribution of CXCL1 to chemoresistance and radioresistance in selected tumors and its influence on the proliferation, migration, and invasion of tumor cells. Additionally, we present the impact of CXCL1 on the microenvironment of reproductive cancers, including its effect on angiogenesis, recruitment, and function of cancer-associated cells (macrophages, neutrophils, MDSC, and T_reg_). The article concludes by summarizing the significance of introducing drugs targeting CXCL1. This paper also discusses the significance of ACKR1/DARC in reproductive cancers.

## 1. Introduction

Reproductive cancers are malignancies that arise in the reproductive organs [1]. The most common reproductive cancer in men is prostate cancer, while in women they are breast cancer, cervical cancer, endometrial cancer, and ovarian cancer. These cancers account for over 26% of all cancer diagnoses, with more than 5 million new cases being diagnosed each year. Moreover, reproductive cancers cause over 1.7 million deaths, representing more than 17% of all cancer-related deaths [2]. Due to their high incidence and mortality, reproductive cancers have been the subject of extensive research.

In recent years, there have been significant changes in the models used to study cancer, with the tumor microenvironment becoming a focus of intense investigation [3]. The research includes not just the interaction between cancer cells but also the interplay between cancer cells and tumor-associated cells that are not cancerous.

Chemokines are one of the components of the tumor microenvironment [4]. Chemokines are chemotactic cytokines that, in the immune system, are responsible for the migration and infiltration of tissues by immune cells. As immune cells constitute an important element of the tumor microenvironment [3], chemokines play a significant role in cancer processes [5].

Chemokines have been divided into four subfamilies based on the conserved cysteine motif at the N-terminus [4]. One of the CXC chemokines with a CXC motif at the N-terminus has 16 representatives in humans: C-X-C motif chemokine ligand 1 (CXCL1) to CXCL17, except for CXCL15. These chemokines can be further divided based on their ability to activate specific chemokine receptors. CXC chemokines that activate C-X-C motif chemokine receptor 2 (CXCR2) include CXCL1 to CXCL8, but not CXCL4. CXCL1 was first described in the 1980s [6]; its first described property was its autocrine stimulation of melanoma cell proliferation, hence its original name: melanoma growth-stimulatory activity (MGSA) [6].

CXCL1 is a chemokine whose main receptor is CXCR2 [7]. However, this chemokine can also activate CXCR1, but the half maximal effective concentration (EC_50_) is two orders of magnitude worse than that for CXCR2 [7]. Another receptor for CXCL1 is atypical chemokine receptor 1 (ACKR1)/Duffy antigen receptor for chemokines (DARC) [8]. The role of this receptor in CXCL1 function is unclear.

The activation of CXCR2 by CXCL1 leads to signal transduction via Gα_i_ [9,10]. Multiple proteins interact with this receptor, resulting in signal transduction independent of G proteins [11]. CXCR2 activation causes cell migration [12] and can also increase proliferation [13]. CXCR2 expression is found in neutrophils [4], making CXCL1 a chemoattractant for these cells. This is its most important physiological property. In many diseases such as multiple sclerosis [14], viral encephalitis [15], periodontitis [16], and alcohol-induced liver injury [17], CXCL1 levels increase, resulting in tissue infiltration by neutrophils. These cells then participate in disease processes, including damage to healthy tissue. In chronic inflammatory states, CXCL1 also participates in the recruitment of MDSCs, leading to a decreased immune response [18]. CXCR2 expression is also found on endothelial cells [19], giving CXCL1 proangiogenic properties. These physiological and non-cancer disease-related properties reflect the role of CXCL1 in cancer processes (Figure 1).

It has been shown that CXCL1 plays a significant role in various cancer-related processes. However, a comprehensive review summarizing all the available knowledge about this chemokine is lacking. Given the vast amount of information available, this review focuses on the role of CXCL1 in cancer-related processes and its association with clinical aspects of reproductive cancers such as breast cancer, cervical cancer, endometrial cancer, ovarian cancer, and prostate cancer.

## 2. Breast Cancer

Female breast cancer is the most commonly diagnosed cancer. In 2020, 2.26 million new cases of this cancer were diagnosed, accounting for 11.7% of all cancer diagnoses [2]. It also caused 685,000 deaths, which accounted for 6.9% of deaths caused by all cancers. Men can also have breast cancer, although it is more than 100 times less common than in women [20]. For this reason, this subsection will only discuss the role of CXCL1 in female breast cancer. Breast cancer is divided into subtypes based on the presence or loss of the expression of estrogen receptor (ER), progesterone receptor (PR), and human epidermal growth factor receptor 2 (HER2) [21,22,23,24,25]. The following main subtypes of breast cancer exist:Luminal A (ER^+^PR^+^HER2^−^);Luminal B (ER^+^HER2^−^ + PR^−^ or Ki-67^high^);Luminal HER2-positive (HER2^+^ + ER^+^ or PR^+^);Non-luminal HER2-positive (ER^−^PR^−^HER2^+^);Triple-negative breast cancer (ER^−^PR^−^HER2^−^).

Triple-negative breast cancer is sometimes referred to as basal-like breast cancer, which is inaccurate because 1/3 of triple-negative breast cancer cases are not basal-like breast cancers [26]. The median age of breast cancer incidence in women is 61 [27]. Risk factors include obesity and an unhealthy diet, and factors that reduce the likelihood of the disease are menopause, a large number of children, and prior breastfeeding. Genetic factors are also important risk factors, accounting for 5–10% of breast cancer cases [24,27]. These include mutations in the breast cancer type 1 and 2 susceptibility gene (*BRCA1* and *BRCA2*); a mutation in one of these genes is associated with a 55–70% probability of developing breast cancer [27].

The action of sex hormones plays an important role in the development of breast cancer—no less important are cytokines, including CXCL1. A bioinformatics study indicated that *CXCL1* is an important gene in breast cancer processes in young adults [28]. There are reports that CXCL1 expression in breast tumors is lower than in normal tissue [29,30,31,32], although some studies show that CXCL1 expression in breast tumors does not differ from healthy tissue [33]. This indicates that CXCL1 may not play an important role in all cases of breast cancer. Other CXCR2 ligands may play a more important role in breast cancer than CXCL1, for example, CXCL5/epithelial-cell-derived neutrophil activating factor 78 (ENA78) and CXCL8/interleukin-8 (IL-8), whose expression is increased in breast tumors [33]. That result was obtained in an analysis of all breast tumors without distinguishing their subtypes.

CXCL1 expression is the lowest in luminal breast cancer and at the same time lower than in normal tissue [31]. In contrast, CXCL1 expression is the highest in basal-like breast cancer and higher than in normal tissue [34]. CXCL1 expression is also increased relative to normal tissue in mesenchymal-like triple-negative breast cancer and basal-like triple-negative breast cancer [35]. CXCL1 expression is higher in inflammatory breast cancer than in other breast cancer subtypes [36]. Inflammatory breast cancer is a rare breast cancer characterized by intense inflammatory responses in the breast. CXCL1 expression in stroma breast cancer is also higher in invasive ductal carcinoma than in ductal carcinoma in situ, which indicates that CXCL1 expression increases with tumor growth [37].

Another study showed that expression in breast cancer decreases with increasing tumor grade [31,32], while another showed that CXCL1 expression does not differ between grade I and III breast cancers [29]. CXCL1 expression is higher in breast cancer metastasis (a group composed mainly of liver, lung, skin, and other metastases) than in primary tumors and higher than in normal tissue [29,36]. Circulating levels of CXCL1 are increased in breast cancer patients compared to healthy subjects [38].

CXCL1 expression in breast cancer cells is lineage dependent. MCF-7 cells (ER^+^PR^+^HER2^−^) produce very small amounts of CXCL1 and there is no autocrine stimulation of proliferation by this chemokine in these cells [39,40]. The highest production of CXCL1 of all breast cancer lines occurs in triple-negative MDA-MB-231 breast cancer cells (ER^−^PR^−^HER2^−^) [29]. This chemokine is produced in 14 to 330 times higher amounts by triple-negative MDA-MB-231 breast cancer cells than by MCF-7 cells [29,40]. On the other hand, high CXCL1 expression does not always occur in triple-negative breast cancer cell lines. An example of this is triple-negative BT-20 cells, which do not secrete CXCL1 in detectable amounts [29].

The expression of the receptor for CXCL1, i.e., CXCR2, is lower in triple-negative breast cancer compared to other breast cancer subtypes [41]. CXCR2 expression is also reduced in ER-negative and HER2-negative breast cancers, indicating a decrease in CXCL1 activity in these tumors and in tumor cells that have lost the expression of these receptors.

The frequency of CXCL1 amplification increases with the progression of breast cancer [42]. It is estimated that 7.5% of primary breast tumors have *CXCL1* gene amplification, while in lymph node and lung metastases, the percentage is around 20% [42]. Furthermore, analyses conducted using the cBioPortal tool (http://www.cbioportal.org/ accessed on 30 March 2023) [43,44] based on TCGA PanCancer Atlas studies [45,46,47,48,49,50] showed that *CXCL1* gene amplification was observed in 1.96% (21/1072) of breast invasive carcinoma cases. In tumors with *PT53* gene mutation, the percentage of *CXCL1* gene amplification is 2.88% (10/347). Additionally, in breast cancer with *BRCA1* or *BRCA2* gene mutations, the amplification percentage of *CXCL1* gene is 3.85% (2/52). When it comes to subtypes of breast cancer, *CXCL1* gene amplification was observed in 1.20% (6/499) of luminal A breast cancer cases, 3.55% (7/197) of luminal B breast cancer cases, 7.69% (6/78) of HER2-positive breast cancer cases, and 1.17% (2/171) of basal breast cancer cases.

The expression of CXCL1 is dependent on GATA-binding protein 3 (GATA3), as demonstrated by experiments on T47D cells [51]. Decreased GATA3 expression leads to increased CXCL1 expression, although the exact mechanism of this relationship is not yet fully understood. It is believed to be associated with the formation of a BRCA1-GATA3 complex, which decreases the expression of *CXCL1* by binding to its promoter [51].

Another factor that increases CXCL1 expression is obesity. BMI is positively correlated with CXCL1 expression levels in breast tumors [52]. Due to the significant influence of CXCL1 in tumorigenesis, obesity in breast cancer patients is associated with a worse prognosis.

Other factors are also responsible for CXCL1 expression in breast cancer. The high expression of CXCL1 in triple-negative MDA-MB-231 cells is partly due to decreased expression of the nonreceptor protein tyrosine kinase Syk. This kinase inhibits nuclear factor κB (NF-κB) activation and thus downregulates CXCL1 expression in MCF-7 cells and other breast cancer cell lines [40]. Other proteins also affect CXCL1 expression in breast cancer. One example is the DEK protein, which downregulates CXCL1 expression [53], and another is SET domain-containing 2 histone lysine methyltransferase (SETD2), a histone methyltransferase that causes histone methylation in the *CXCL1* gene promoter [54]. In breast tumors, SETD2 expression is downregulated compared to healthy tissue [55], and thus CXCL1 expression is upregulated [54]. Another factor that decreases CXCL1 expression in breast cancer cells is either decreased expression or loss of HER2 expression [56]. Additionally, a mutation in *KRAS* leads to the activation of mitogen-activated protein kinase (MAPK) cascades, which then leads to increased CXCL1 expression in breast cancer cells [57].

CXCL1 expression in breast tumors is also significantly influenced by intercellular signaling and secretory factors. High CXCL1 expression in triple-negative breast cancer may be due to transforming growth factor-α (TGF-α) [35]. It is an epidermal growth factor receptor (EGFR) ligand. At the same time, TGF-α does not affect the expression of CXCR2 ligands in non-triple-negative breast cancer cells. CXCL1 expression in breast cancer cells is also increased by secretory factors such as prostaglandin E_2_ (PGE_2_) [58], interleukin-17 (IL-17) derived from T helper type 17 (Th17) cells [59,60], loss of response to transforming growth factor-β (TGF-β) [61,62], tumor necrosis factor-α (TNF-α) [42], leukemia inhibitory factor (LIF) [63], plasminogen activator inhibitor-1 (PAI-1) [64], and Notch signaling [65].

CXCL1 expression in breast cancer cells is downregulated by TGF-β [61,66]. In particular, TGF-β prevents the induction of CXCL1 expression by IL-17 [61]. At the same time, the expression of the receptor for TGF-β, i.e., TβRII, is downregulated in breast tumors [62], resulting in an increase in CXCL1 expression in the tumor. This is followed by the recruitment of myeloid-derived suppressor cells (MDSC) into the tumor niche—these cells secrete TGF-β and interleukin-6 (IL-6), which results in the differentiation of CD4^+^ T cells into Th17 cells that secrete IL-17, a mechanism further increasing CXCL1 expression in breast cancer cells [59,60,61].

The effect of TNF-α on CXCL1 expression is important in resistance to chemotherapy. Anticancer drugs increase TNF-α expression, which results in the activation of the immune system and the destruction of breast cancer cells [42]. At the same time, TNF-α also increases the expression of CXCL1, which results in the recruitment of MDSC, cells that inhibit the antitumor immune system response—this means that CXCL1 in this situation is responsible for chemoresistance [42].

CXCL1 expression is also upregulated by LIF secreted by cancer-associated adipocytes [63]. LIF expression in these cells is increased as a result of signal transducer and activator of transcription 3 (STAT3) activation by CXCL1 from breast cancer cells. Therefore, a positive feedback loop is formed between cancer-associated adipocytes and breast cancer cells; CXCL1 is one of the components of the loop.

CXCL1 expression in breast cancer cells can also be altered by the plasminogen activation regulatory system. A study on mouse 4T1 breast cancer cells showed that urokinase-type plasminogen activator (uPA) decreases, while PAI-1 increases, the expression of CXCR2 ligands [64]. CXCL1 expression in breast cancer cells is also increased by Notch signaling [65]. The activation of this pathway is followed by the activation of hairy/enhancer-of-split related with YRPW motif-like (HEYL), a protein that attaches to the *CXCL1* promoter, thus increasing the expression of CXCL1. In this way, the expression of CXCL2/growth-regulated oncogene-β (GRO-β) and CXCL3/growth-regulated oncogene-γ (GRO-γ) also increases in breast cancer cells [65].

In breast tumors, cancer-associated fibroblast (CAF) may be the main source of CXCL1 [37,67,68,69]. CXCL1 expression in CAF is increased under the influence of basal-like breast cancer cells [70], while luminal breast cancer cells do not increase CXCL1 expression in CAF. This may be the reason for the higher CXCL1 expression in basal-like triple-negative breast cancer than in other breast cancer subtypes [34,35].

CXCL1 expression in CAF is also associated with a decrease in TGF-β levels in breast tumors. This cytokine reduces CXCL1 expression in CAF [37,69]. Decreased levels of TGF-β increase the expression of hepatocyte growth factor (HGF) [68], a growth factor that, through its receptor c-Met, increases the expression of CXCR2 ligands in CAF, as shown by experiments on mouse cells [68].

Oncostatin M, which is secreted by tumor-associated neutrophils (TAN) and tumor-associated macrophages (TAM) [71,72], is also responsible for the increase in CXCL1 expression in CAF. On the other hand, oncostatin M also increases the expression of other factors in CAF, such as vascular endothelial growth factor (VEGF), CC motif chemokine ligand (CCL)2/monocyte chemoattractant protein 1 (MCP-1), CCL5/regulated on activation, normally T-cell-expressed and -secreted (RANTES), CCL19/EBI1-ligand chemokine (ELC), CCL21/secondary lymphoid tissue chemokine (SLC), and CXCL12/stromal-derived factor-1 (SDF-1) [72].

TAM are also responsible for CXCL1 production in breast tumors [34,73,74,75]. The amount of CXCL1 secreted by TAM may be the highest of all cytokines produced by these cells, as shown by studies of TAM isolated from lung metastasis from a mouse model [34].

Another source of CXCL1 in breast tumors may be adipose-derived mesenchymal stem cells (MSC) [76], which increase CXCL1 expression under the influence of interleukin-1β (IL-1β) from breast cancer cells [77]. Importantly, adipose-derived MSC also secrete CXCL8/IL-8. However, these cells are important in breast tumor development only in some models—they support the development of tumors arising from MCF-7 and ZR-75-30 lines but not from MDA-MB-231 lines [76].

CXCL1 is involved in tumorigenesis in breast cancer. This effect depends on whether a given cell line expresses the CXCL1 receptor, i.e., CXCR2 [29]. CXCL1 increases the proliferation of breast cancer cells, as has been shown in experiments on triple-negative MDA-MB-231 breast cancer lines, triple-negative HCC-1937 breast cancer cells, HER2-negative MCF-7 cells, and HER2-positive SKBR3 breast cancer cells (ER^−^PR^−^HER2^+^) [78]. In triple-negative breast cancer cells, there is often an autocrine increase in cell proliferation by CXCL1. This chemokine is produced and secreted by these cells, which then increases their proliferation [78].

CXCL1 is also important in the function of breast cancer stem cells. These cells express CXCL1 as well as the receptor for this chemokine, i.e., CXCR2 [79]. CXCR2 expression in breast tumors is mainly found in cancer stem cells [41], and for this reason, CXCR2 can be considered a marker of breast cancer stem cells. The simultaneous expression of CXCL1 and the receptor for this chemokine, CXCR2, allows breast cancer stem cells to act on themselves in an autocrine manner. This results in the proliferation of these cells, which increases their stemness and causes self-renewal [41,79], as shown by experiments on triple-negative MDA-MB-231 breast cancer [73,74].

CXCL1 has pro-survival and anti-apoptotic effects on breast cancer cells. CXCL1 increases the expression of anti-apoptotic Bcl-2 family proteins and decreases the expression of pro-apoptotic Bcl-2 family proteins [80], which is important in chemoresistance and radioresistance.

CXCL1 also causes the migration and epithelial-to-mesenchymal transition (EMT) of breast cancer cells, with the effect being lineage-dependent. CXCL1 causes the migration of triple-negative MDA-MB-231 breast cancer cells [40,73,78], MCF-7 cells (ER^+^PR^+^HER2^−^) [78,81], Zr-75-1 cells (ER^+^PR^+^HER2^+^) [81], triple-negative HCC-1937 breast cancer cells, and HER2-positive SKBR3 breast cancer cells (ER^−^PR^−^HER2^+^) [78]. On the other hand, CXCL1 does not cause the migration of T47D cells (ER^+^PR^+^HER2^−^) [81], even though T47D cells express CXCR2 [29]. CXCL1 induces breast cancer cell migration through the activation of extracellular signal-regulated kinase (ERK) MAPK, which increases the expression of matrix metalloproteinase 2 (MMP2) and matrix metalloproteinase 9 (MMP9) [82]. Also important in the induction of breast cancer cell migration by CXCL1 are Akt/protein kinase B (PKB), NF-κB, and STAT3 [69].

CXCL1 also induces the EMT of breast cancer cells, as shown by experiments on MDA-MB-231, HCC-1937, SKBR3, and MCF-7 cell lines [34,78]. This process is dependent on the CXCL1-induced activation of the NF-κB-sex-determining region Y-related high-mobility group box 4 (SOX4) pathway [34].

CXCL1-induced migration can be either directly via the activation of the CXCR2 receptor on breast cancer cells or indirectly. Indirectly, CXCL1 causes the recruitment of neutrophils into the tumor niche. TAN cause breast cancer cells to migrate by activating intracellular adhesion molecule-1 (ICAM-1) on these cells [83]. TAN also show the expression of β_2_-integrins, which involves the direct interaction of neutrophils with breast cancer cells and the interaction of β_2_-integrins with ICAM-1 on the cancer cell. CXCL1 can also indirectly induce the EMT of breast cancer cells through recruiting MDSC, which then causes the EMT of breast cancer cells via IL-6 [84].

CXCL1 is also important in breast cancer lymph node metastasis. Breast cancer cells cause an increase in CXCL1 expression in lymphatic endothelial cells (LEC) [85], which causes breast cancer cells to migrate into lymphatic vessels, resulting in lymph node metastasis.

CXCL1 may also be important in the formation of breast cancer metastasis in other organs. Blood levels of this chemokine are positively correlated with the number of circulating breast cancer cells [38], and the more circulating cancer cells, the greater the likelihood of metastasis.

CXCL1 has also been found to have a significant effect on the formation of metastasis of breast cancer cells to various organs. CXCL1 may be important in the bone metastasis of the cancer in question. This chemokine acts on osteoclasts precursors through the CXCR2 receptor on these cells [86,87,88,89]. This leads to osteoclast maturation, bone remodeling, and the formation of bone metastasis. At the same time, CXCL8/IL-8 appears to be more important in the formation of bone metastasis of breast cancer than CXCL1 [90].

CXCL1 seems to be involved in the formation of brain metastasis of breast cancer. In the formation of brain metastasis, a feedback loop is formed between breast cancer cells and astrocytes [91]. Breast cancer cells secrete IL-1β, in response to which astrocytes secrete HGF, which in turn increases IL-1β expression in breast cancer cells. IL-1β also increases the expression of CXCL1 and CXCL8/IL-8. These chemokines are responsible for the formation of the perivascular niche and angiogenesis in the initial stages of brain metastasis in breast cancer [91].

CXCL1 may also play some role in the formation of lung metastasis in breast cancer. The circulating levels of CXCL1 are the highest in patients with lung metastasis in breast cancer and higher than in patients with bone metastasis [38]. CXCL1 causes an increase in the adhesion of circulating breast cancer cells to human pulmonary microvascular endothelial cells, that is, to the walls of blood vessels in the lungs [92]. In lung fibroblasts, CXCL1, as well as other CXCR2 ligands, increases the expression of CCL2/MCP-1 and CCL7/MCP-3, both of which increase cholesterol synthesis in breast cancer cells in the lung. This leads to angiogenesis and the generation of lung metastasis in breast cancer [93]. Notably, this mechanism occurs in triple-negative breast cancer.

CXCL1 acts on tumor-associated cells. CXCL1 in breast tumors causes angiogenesis by acting directly on endothelial cells [58,76]. CXCL1 can also indirectly cause angiogenesis by inducing an increase in VEGF expression in breast cancer cells [78].

CXCL1 is involved in the recruitment of various cells to the tumor niche. It causes the recruitment of neutrophils [57,83,94], granulocytic-myeloid-derived suppressor cells (G-MDSC) [52,57], and breast resident adipose tissue-derived MSC [95]. This process may also be important for the location of the recruited cells in the tumor. Breast cancer cells secrete CXCL1 and CXCL8/IL-8, resulting in the chemotaxis of neutrophils into breast cancer cells [83]. Direct contact between these cells is then possible, resulting in the stimulation of breast cancer cell migration. This process is dependent on ICAM-1 on breast cancer cells and β_2_-integrins on neutrophils [83].

CXCL1 can recruit c-Kit^+^Ly6a/Sca1^+^ hematopoietic stem/progenitor cells into the tumor niche, as demonstrated by experiments in mice [96]. These cells differentiate into MDSC. CXCL1 also induces the recruitment of naive CD4^+^ T cells, which are transformed into regulatory T (T_reg_) cells under the influence of the tumor microenvironment [75]. At the same time, CXCL1 can induce the differentiation of naive CD4^+^ T cells into T_reg_. CXCL1 also acts on cancer-associated adipocytes, causing STAT3 activation in them, which leads to the production of LIF [63] which in turn increases CXCL1 production in breast cancer cells.

CXCL1 may be involved in chemoresistance and radioresistance. The expression of this chemokine in breast cancer cells is increased by ionizing radiation in microbeam radiotherapy [97], as well as by chemotherapeutics such as paclitaxel [98] and doxorubicin [99]. At least in the case of doxorubicin, this is dependent on a defect in p53 in breast cancer cells and the activation of NF-κB in these cells. The chemotherapeutic-induced increase in CXCL1 expression may be responsible for the increase in TNF-α expression [42]. CXCL1 also induces the recruitment of MDSC, cells that inhibit an anticancer immune response produced by radio- and chemotherapy (Figure 2). CXCL1 exerts its anti-apoptotic effect by affecting the expression of Bcl-2 family proteins [80], which inhibits breast cancer cell apoptosis induced by anticancer therapy.

High CXCL1 expression in breast tumors is positively correlated with lymph node metastasis [34,100] and TNM stage [34]. On the other hand, there is a study showing that stromal CXCL1 expression is not correlated with lymph node metastasis [37]. The level of CXCL1 expression is also not related to tumor size [34,37,100] or histologic grade [34]. The level of CXCL1 in the tumor is not related to the age of patients [37]. Its highest expression is found in basal-like breast cancer [34] and triple-negative breast cancer [35].

Depending on the literature source, the correlation of CXCL1 expression with negative PR and ER status may vary. CXCL1 can be correlated with negative PR [100] and negative ER status [82,94,100], and in some studies, it is not correlated [34,37]. Circulating levels of CXCL1 are not associated with PR and ER status in breast tumors [38]. Additionally, CXCL1 expression is not associated with HER2-negative status in breast tumors [34,37,100].

Considering all breast cancer cases, the association between the level of CXCL1 expression in the tumor and prognosis varies depending on the literature source (Table 1). High CXCL1 expression in breast tumors may be associated with either a worse [34,37,100] or better prognosis [30,32]. The level of CXCL1 in the blood may be associated with the prognosis of breast cancer patients, with a higher circulating level of CXCL1 indicating a worse prognosis [38]. Conversely, higher CXCL1 expression is associated with a worse prognosis for patients with basal breast cancer [52]. For ERα-positive breast cancer, depending on the work cited, high CXCL1 expression in the tumor may be associated with either a worse [29] or better prognosis [31].

## 3. Cervical Cancer

In 2020, 604 thousand new cases of cervical cancer accounted for 3.1% of all cancers [2]. The nearly 342 thousand deaths caused by this cancer accounted for 3.4% of deaths caused by all cancer-related deaths. Its most important risk factor is infection with the oncogenic subtypes of human papillomavirus (HPV) [101], hence the significance of vaccination against this virus.

CXCL1 may be important in the development of cervical cancer in pre-cancerous cervical lesions. The expression of this chemokine along with other CXCR2 ligands, including CXCL7/neutrophil-activating protein 2 (NAP-2) and CXCL8/IL-8, is significantly elevated in cervical intraepithelial neoplasia grades 1 (CIN1) and CIN2 [102]. CXCL1 expression is also slightly elevated in CIN3 compared to healthy individuals [102]. In further stages of tumorigenesis, CXCL1 levels are also increased. CXCL1 expression is higher in cervical tumors relative to healthy tissue [103,104]. Additionally, CXCL1 levels in the serum of cervical cancer patients are higher than in healthy subjects [105].

Using cBioPortal (http://www.cbioportal.org/ accessed on 30 March 2023) [43,44], which analyzed TCGA PanCancer Atlas studies [45,46,47,48,49,50], it was found that the amplification of the *CXCL1* gene occurs in 0.67% (2/297) of cases of cervical squamous cell carcinoma, while deletion occurs in 0.34% (1/297) of cases.

CXCL1 is involved in tumorigenic processes. It increases the proliferation and inhibits the apoptosis of cervical cancer cells [104,106,107]. This effect depends on ERK MAPK activation [104]. CXCL1 has an autocrine effect on these cells, which is associated with high basal NF-κB activation caused by the overexpression of A-kinase-interacting protein 1 (AKIP1) [106]. NF-κB attaches to the CXCL1 promoter, increasing the expression of CXCL1, which is produced and secreted by cervical cancer cells and then acts on these cells. In the absence of this autocrine loop, the apoptosis of the cells in question occurs [107]. CXCL1 also causes the migration of the cancer cells of the cancer in question [104,107]—this effect depends on ERK MAPK activation [104]. CXCL1 also acts on cells in the tumor niche, causing angiogenesis [106].

CXCL1 is associated with cervical tumor growth. Therefore, the expression of this chemokine is positively correlated with tumor size [108]. It has been observed that the level of CXCL1 in the serum of patients with squamous cell cervical cancer in stage II and stage III is higher than in stage I [109]. At the same time, CXCL1 expression is not associated with HPV status and lymph node metastasis [109]. The great significance of CXCL1 in tumorigenesis is confirmed by the correlation between patient prognosis and the level of CXCL1 expression in cervical tumors (Table 2). The higher the expression of the chemokine, the worse the prognosis for patients with cervical cancer [103,104,108].

## 4. Endometrial Cancer

In 2020, 417 thousand new cases of endometrial cancer were diagnosed, which accounted for 2.2% of all cancers [2]. There were also nearly 97,000 deaths caused by this cancer—1.0% of all cancer-related deaths. A major risk factor for endometrial cancer is estrogen exposure [111]. The role of CXCL1 in this cancer has been poorly studied; nevertheless, available data suggest that this chemokine may be significant in tumorigenic processes in this cancer.

Studies using qRT-PCR on endometrial adenocarcinoma tissue have shown that CXCL1 expression is elevated in these tumors relative to healthy tissue [112]. Additionally, according to the GEPIA portal [110], which compares data from the Cancer Genome Atlas (TCGA) [113] with data from normal tissues from GTEx [114,115], CXCL1 expression is upregulated in uterine corpus endometrial carcinoma. Moreover, CXCL1 levels in the serum of patients with endometrial cancer are higher than in healthy subjects [105].

Using cBioPortal (http://www.cbioportal.org/ accessed on 30 March 2023) [43,44], which analyzed TCGA PanCancer Atlas studies [45,46,47,48,49,50], it was found that there is no amplification of the *CXCL1* gene in a sample of 529 cases of uterine corpus endometrial carcinoma.

CXCL1 expression in the endometrial tumor is increased by prostaglandin F_2α_ (PGF_2α_) [112]; this process is dependent on EGFR and ERK MAPK [112]. Progesterone and calcitriol decrease CXCL1 expression in endometrial cancer cells by decreasing NF-κB activation [116].

CXCL1 is crucial for tumor cell invasiveness in endometrial cancer [116]. It also causes the recruitment of neutrophils into the tumor niche [112].

According to the GEPIA portal, the expressions of CXCL1 do not affect the prognosis for patients with uterine corpus endometrial carcinoma (Table 3) [110].

## 5. Ovarian Cancer

In 2020, nearly 314 thousand new cases of ovarian cancer were diagnosed, a number which accounted for 1.6% of all new cancer cases that year [2]. Also in 2020, 207 thousand deaths were caused by this cancer, which accounted for 2.1% of all cancer-related deaths. The median age of onset is about 60 years, i.e., in the post-menopausal period [117]. Factors that reduce the incidence of ovarian cancer include having a history of pregnancy and previous use of oral contraception [117]. In contrast, factors that increase the likelihood of developing ovarian cancer are mutations in the *BRCA1* and *BRCA2* genes [118,119]. The characteristic feature of this cancer is abdomen metastasis [117].

CXCL1 expression is elevated in ovarian cancer [120,121]. Although CXCL1 expression is found in a significant number of ovarian carcinoma cases [122], its role can be largely played by CXCL8/IL-8, whose expression is found in all cases of ovarian carcinomas [122].

Blood levels of CXCL1 in ovarian cancer patients are higher than in healthy individuals [105,123,124]. For this reason, both CXCL1 and CCL18 may be used as biomarkers in the diagnosis of ovarian cancer [123].

CXCL1 levels in cyst fluid in ovarian cancer patients are higher than in serum [125]. Nevertheless, CXCL1 may be important only in the early stages of ovarian tumor development [125,126]. The pre-spreading of ovarian tumors shows the expression of CXCL1 and other CXCR2 ligands as well as the expression of CXCR4 [126]. The post-spreading of ovarian tumors shows high expression of CCL20/liver and activation-regulated chemokine (LARC), CXCL17, and CXCR4 [126]. This is associated with an increase in CCL20/LARC expression by CXCL1 [126], indicating some change in the expression of secretory factors during ovarian cancer development, in which CXCL1 is crucial at its beginning and CCL20/LARC at later stages.

Using cBioPortal (http://www.cbioportal.org/ accessed on 30 March 2023) [43,44], which analyzed TCGA PanCancer Atlas studies [45,46,47,48,49,50], it was found that the amplification of the *CXCL1* gene occurs in 2.57% (15/583) of cases of ovarian serous cystadenocarcinoma, while deletion occurs in 0.34% (2/583) of cases.

CXCL1 may play a significant role in the tumorigenesis and formation of ovarian cancer, as confirmed by studies on *CXCL1* gene polymorphisms. A study on the Chinese Han population showed that the rs11547681 variant affects the risk of ovarian cancer [127].

High fertility is a factor that reduces the risk of the disease, and it is associated with reduced aggressiveness of ovarian cancer [128]. A study on mice has shown that a carried pregnancy during the reproductive period alters the physiology of the omental fat band in the postmenopausal period. This is followed by a lower level of CXCR2 ligand expression in the omental fat band compared to females who have never been pregnant [129]. This reduces the likelihood of developing metastasis of ovarian cancer in females who have had term pregnancies [129,130].

Another reason is a change in the expression of other proteins and microRNAs affecting the expression of the chemokine in question. In ovarian tumors, there is a reduction in the expression of miR-27b-5p relative to healthy tissue [131]. This is a microRNA that directly decreases CXCL1 expression; hence, a decrease in its expression increases CXCL1 expression.

Mutations in the *TP53* gene also lead to increased CXCL1 expression in ovarian cancer [132], which is associated with increased NF-κB activation [132], as well as the direct attachment of p53 with a gain-of-function mutation to the *CXCL1* promoter [133].

In ovarian cancer cells, CXCL1 expression is also increased by Snail [124], a transcription factor important in EMT. It increases CXCL1 expression by directly attaching to the promoter of this chemokine and by causing an increase in NF-κB activation.

Another protein that increases NF-κB activation in ovarian cancer cells is scaffold adapter GRB2-associated binding protein 2 (GAB2), whose expression is elevated in ovarian tumors [134]. An increase in NF-κB activation results in the attachment of this transcription factor to the *CXCL1* promoter, which increases the expression of this chemokine.

CXCL1 expression in ovarian cancer cells was also found to be dependent on metastasis-associated gene 1 (MTA1), the expression of which is increased in ovarian tumors [135].

CXCL1 expression in cancer cells is also elevated by secretory factors such as epidermal growth factor (EGF) [136,137], IL-17 [138], protease-activated receptor 1 (PAR1) [139], and TNF-α [137,140]. TNF-α levels are elevated in ovarian cancer [141] and depend on the upregulation of TNF-α expression by TNF-α. TNF-α expression is also increased by lysophosphatidic acid (LPA) [140]. Importantly, the increase in CXCL1 expression by either EGF or TNF-α depends on the lineage of the ovarian cancer. In some lines, both factors increase CXCL1 expression, in some, only one factor, and in some, neither factor increases CXCL1 expression [137].

By activating CXCR2, CXCL1 increases the proliferation of ovarian cancer cells [134,142,143,144,145]. This effect may be dependent on non-cancerous CXCL1-producing cells: from CAF, a parent tumor [146] and omental human peritoneal mesothelial cells at the beginning of metastasis formation [144]. The magnitude of the effect on ovarian cancer cell proliferation depends on the status of p53 in these cells. CXCL1 increases the proliferation of ovarian cancer cells with functional p53 much more than with either mutated *TP53* or the deletion of this gene [145], which is related to the activation of murine double minute 2 (Mdm2) by CXCR2. Mdm2 inhibits p53 activity by the ubiquitination of this protein, leading to the proteolytic degradation of p53 [147]. CXCL1 causes anchorage-independent growth [134].

The enhancement of ovarian cancer cell proliferation by CXCL1 may occur through EGFR transactivation [142]. CXCR2 activation causes the proteolytic cleavage of heparin-binding EGF-like growth factor (HB-EGF), an EGFR ligand. This process is MMP-dependent, as confirmed by the use of pan-MMP inhibitor GM6001.

CXCL1 exerts its anti-apoptotic effects by acting on Bcl-2 family proteins [143].

CXCL1 is also important in the migration and EMT of ovarian cancer cells [116,133], which is related to the increased expression of MMP2 and MMP9 by CXCL1 [116]. CXCL1 may play a significant role in ovarian cancer metastasis. The expression of CXCR2 ligands occurs in omental milky spots, adipose tissue, and blood vessels, as shown by experiments in mice [130]. The expression of CXCR2 ligands is increased in ovarian cancer cells by secretory factors secreted by adipocytes [148]. Ovarian cancer cells also cause an increase in CXCL1 and CXCL8/IL-8 expression in omental human peritoneal mesothelial cells [144]. These ligands, similar to CXCR2, are important in the formation of intraperitoneal cavity metastasis in ovarian cancer. Specifically, they stimulate proliferation and have a pro-survival effect on ovarian cancer cells after migration from the parent tumor [144].

CXCL1 is also important in the function of tumor-associated cells. CXCL1, secreted by ovarian cancer cells, causes the senescence of fibroblasts [149] in a process dependent on functional p53 in fibroblasts. The senescence of fibroblasts leads to an increase in CXCL1 expression in these cells [146,149]. For this reason, CAF show CXCL1 expression in the ovarian cancer niche [67,146]. Also in ovarian cancer, CAF secrete other secretory factors, including VEGF, IL-6, granulocyte colony-stimulating factor (G-CSF), CCL2/MCP-1, and CXCL8/IL-8 [146], which are involved in tumorigenesis and lead to cancerous tumor growth.

MSC may also be responsible for CXCL1 expression in ovarian tumors [150]. The expression of this chemokine in these cells is increased by chemotherapeutics including carboplatin, leading to carboplatin resistance. CXCL1 also acts on G-MDSC, causing their recruitment to the tumor niche [124].

CXCL1 also causes angiogenesis in ovarian tumors [134].

CXCL1 levels in ovarian tumors are not stage-related [120,121]. The level of CXCL1 expression in ovarian tumors differentially affects the prognosis for patients, according to the cited work. The level of CXCL1 expression in ovarian tumors may be associated with a better prognosis [120], worse [124], or no effect on prognosis [121].

High serum CXCL1 levels are associated with a worse prognosis for ovarian cancer patients (Table 4) [124]. This may be related to resistance to chemotherapy. Higher serum CXCL1 levels are a marker of resistance to carboplatin in patients with high-grade serous ovarian carcinoma (HGSOC) [151] and ovarian cancer [150]. It has been observed that carboplatin increases CXCL1 expression in cancer-associated MSC, resulting in increased resistance to subsequent cycles of chemotherapy. This is an important mechanism of chemoresistance [150].

## 6. Prostate Cancer

Prostate cancer is diagnosed only in men. Risk factors for this cancer include high age, poor diet (fatty food with little fish), obesity, sexually transmitted diseases (syphilis, gonorrhea infections, human papillomavirus), and genetic factors [152]. When screening for prostate cancer, prostate-specific antigen (PSA) levels in the blood are tested [153]. Prostate cancer ranks fourth in terms of the number of diagnoses among cancers [2]. It is estimated that more than 1.41 million new cases are diagnosed annually, accounting for 7.3% of all cancer diagnoses [2]. Additionally, it results in 375 thousand deaths per year, which accounts for 3.8% of deaths caused by all cancers [2].

CXCL1 expression may be upregulated in prostate tumors [154], but other available studies show either no difference in the expression of this chemokine in prostate cancer [155] or in serum levels in prostate cancer patients relative to healthy subjects [156]. The expression of this chemokine in prostate cancer may depend on the patients studied. Obese patients have high CXCL1 expression in prostate tumors [157]; in contrast, lean patients rarely have high CXCL1 expression in these tumors.

Using cBioPortal (http://www.cbioportal.org/ accessed on 30 March 2023) [43,44], which analyzed TCGA PanCancer Atlas studies [45,46,47,48,49,50], it was found that the amplification of the *CXCL1* gene occurs in 0.61% (3/494) of cases of prostate adenocarcinoma, while deletion occurs in 0.20% (1/494) of cases.

CXCL1 may be an important driver of prostate cancer. In older individuals, there is an increase in the expression of CXCR2 ligands, including CXCL1, in prostate fibroblasts [158], which leads to cell proliferation in the prostate, particularly of fibroblasts and epithelial cells. The resulting benign prostatic hypertrophy can signal the beginning of prostate cancer. CXCL1 also plays an important role in the further stages of tumorigenesis. Prostatic intraepithelial neoplasia (PIN) is characterized by inflammation and is associated with an increase in CXCR2 expression compared to healthy tissue [159] as well as an increase in CCL2/MCP-1 levels, which causes the recruitment of macrophages [160]. Macrophages secrete CXCL1, which causes cell proliferation in PIN [160]. Another factor that increases CXCL1 expression in PIN is IL-17 [161].

The aforementioned activity of CXCL1 in PIN leads to the transformation of PIN into prostate cancer, which, in its early stages, is characterized by inflammation. Epithelial cells secrete IL-1α and IL-1β, which results in the increased expression of CXCR2 ligands, including CXCL1, in transformed prostate stromal cells [162]. In contrast, CXCL1 can also inhibit prostate cancer development by reinforcing senescence [159]. That is, this chemokine can inhibit prostate cancer cell proliferation and tumor growth [159]. On the other hand, it can also induce the senescence of tumor-associated cells and induce SASP in them [163,164], which supports the development of prostate cancer.

CXCL1 is produced by prostate cancer cells [165], as well as myofibroblasts [154] and CAF that have lost TGF-β type II receptor (TβRII) expression [166]. The level of this chemokine in prostate tumors is dependent on an autocrine increase in CXCL1 expression [167]. CXCL1 causes the activation of NF-κB, which increases the expression of CXCL1. At the same time, in castration-resistant prostate cancer cells, the expression of CXCL1 is mediated by NF-κB2/p52, whose activity is elevated relative to less aggressive prostate cancers [168].

The importance of individual CXCR2 ligands varies in different lines. Cells of the PC-3 line (isolated from bone metastasis) mainly produce CXCL8/IL-8, while cells of the DU 145 line (isolated from brain metastasis) mainly produce CXCL1 [169]. The expression of CXCR2, the receptor for CXCL1, is higher in less aggressive prostate cancer lines [165]. Importantly, cancer cells in prostate tumors are not a homogeneous population. The majority of those cancer cells are AR^+^CXCR2^−^ luminal cancer cells [170], but there are also AR^−^CXCR2^+^ neuroendocrine cancer cells, which account for about 0.5% of all tumor cells in low-grade prostate cancer to 20% in metastasis prostate cancer [170]. In small-cell neuroendocrine carcinoma (SCNC) of the prostate (a type of prostate cancer), up to 74% are AR^−^CXCR2^+^ neuroendocrine cancer cells. Due to the expression of CXCR2 in these cells, CXCL1 may affect the function of these cells.

CXCL1 shows weak stimulatory properties for prostate cancer cell proliferation, as shown by studies on PC-3 and DU 145 cell lines [171]. It does induce prostate cancer cell migration [154,171,172], with CXCR1 and CXCR2 receptors involved in this process [171]. The activation of CXCL1 receptors is followed by the activation of Src family kinases, which results in EMT [154,171]. Additionally, CXCL1 reduces fibulin-1 expression in prostate cancer cells [171] and causes the activation of NF-κB, which forms a complex with histone deacetylase 1 (HDAC1). This complex reduces the acetylation of histone in NF-κB-binding sites of the fibulin-1 promoter, thus reducing the expression of this protein. As fibulin-1 regulates ECM function, changes resulting from decreased fibulin-1 expression in the ECM facilitate prostate cancer migration.

In bone, CXCL1 facilitates the formation of bone prostate metastasis. The exposure of prostate cancer cells to CXCL1 increases the adhesion of these cells to type-I collagen [166]. This is the ECM found in bone. Therefore, CXCL1 promotes bone metastasis by increasing the number of circulating prostate cancer cells retained in bone. Prostate cancer cells isolated from bone metastasis show the expression of multiple factors including FGF3, FGF19, GDF15, and the described CXCL1 [88]. These factors are involved in the formation of bone metastasis. FGF3 and FGF19 increase osteoblast proliferation, and GDF15 increases osteoblast differentiation. CXCL1, on the other hand, acts on osteoclast precursors, which leads to the differentiation of these cells into osteoclasts [86,89,173] and thus to bone remodeling, which enhances prostate metastasis. Studies on mouse cells have shown that marrow adipocytes under the influence of factors from prostate cancer secrete CXCR2 ligands [87], which indicates that CXCL1 may work via this mechanism, although this needs to be confirmed on human cells. CXCL1 increases osteoclast maturation, which leads to bone remodeling and the forming of prostate metastasis. The number of adipocytes is increased in the bone marrow in older or obese individuals [87], which facilitates the formation of bone prostate metastasis in elderly or obese individuals.

Increased CXCR2 expression on cancer cells transforms these cells into neuroendocrine cancer cells [170]. At the same time, it is unclear whether this effect is due to CXCR2 receptor expression or to the activation of this receptor by CXCL1 or another ligand. Neuroendocrine cancer cells show an increased stemness, and EMT begins to take place in these cells. AR^−^CXCR2^+^ neuroendocrine cancer cells cause the formation of a tumor microenvironment, resulting in the survival of AR^+^ luminal cancer cells during hormone therapy [170]. Additionally, when CXCR2 expression is increased in prostate cancer cells, the expression of secretory factors such as VEGF, angiogenin, thrombopoietin, oncostatin M, IGF-1, CCL2/MCP-1, CCL22/MDC, and M-CSF is increased [170], namely factors that are involved in tumorigenesis.

CXCL1 acts on tumor-associated cells. CXCL1 acts on CXCR2^+^ macrophages in the tumor niche in prostate tumors, which causes stronger M2 polarization of these cells [174,175]. Through these cells, CXCL1 causes angiogenesis and cancer-immune evasion [176]. This chemokine also causes the mobilization of white adipose tissue (WAT) adipose stromal cells (ASC), and can then recruit these cells into the tumor niche [157] and transform them into myofibroblasts (also by other factors), and then stimulate tumor cells to proliferate and have pro-angiogenic effects.

The levels of CXCL1 in prostate tumors are higher in T-stage and N-stage tumors than in earlier stages [177]. Additionally, CXCL1 expression is higher in tumors with a higher Gleason score [178]. However, other available studies have not confirmed an association between CXCL1 levels and the Gleason score [159,177] or the T stage [178]. Nevertheless, higher CXCL1 expression in prostate tumors is associated with a worse prognosis for the patient (Table 5) [154,177], which indicates an important function of CXCL1 in tumorigenic processes in prostate cancer. For this reason, therapeutic approaches targeting the CXCL1-CXCR2 axis are being tested. An example of this is HL2401, which is a monoclonal anti-human CXCL1 antibody [179]. Additionally, SB225002 [180] shows confirmed anticancer properties. It is a dual CXCR1/CXCR2 inhibitor that reduces the effects of not only CXCL1 but also other CXCR2 ligands including CXCL8/IL-8 [181]. The best results can be obtained when therapy is combined with a drug that targets the CXCL1-CXCR2 axis. Hormonal therapy induces an increase in AR^−^CXCR2^+^ neuroendocrine cancer cells in prostate tumors [170] which causes the survival of AR^+^ luminal cancer cells during subsequent doses of hormonal therapy. For this reason, hormonal therapy should be combined with the use of some CXCR2 inhibitors to prevent this mechanism of resistance to treatment. Also in chemotherapy, particularly when using oxaliplatin, CXCL1 expression increases in prostate cancer cells [182] which leads to resistance to treatment. For this reason, some sort of CXCR2 inhibitor should be used along with standard chemotherapy.

## 7. ACKR1 in Reproductive Cancers

CXCR2 is considered the primary receptor for CXCL1 [183], but CXCL1 can also bind to the atypical chemokine receptor 1 (ACKR1)/Duffy antigen receptor for chemokines (DARC) [21]. ACKR1 is known to bind to other chemokines such as CCL1, CCL2, and CCL5, but its exact role in chemokine signaling remains unclear. Some studies have suggested that ACKR1 regulates the availability of certain chemokines, as mice lacking ACKR1 exhibit altered levels of certain chemokines [184]. Additionally, ACKR1 may be involved in transporting chemokines through the endothelium, which is critical for the migration and recruitment of certain immune cells [185].

The overexpression of ACKR1 has been shown to exert an antitumor effect. ACKR1 on tumor cells [186,187] and erythrocytes [188] inhibits angiogenesis by reducing the levels of pro-angiogenic chemokines. This has been found to be particularly relevant for inhibiting tumor growth in breast [186], ovarian cancer [187], and prostate tumors [188]. Another mechanism underlying the antitumor effect of ACKR1 is related to the expression of CD82/KAI1 on tumor cells [189]. The direct interaction of ACKR1 on vascular endothelium with CD82/KAI1 on tumor cells leads to the inhibition of tumor cell proliferation and metastasis. In addition, ACKR1 can counteract the effect of CXCL1 through its interaction with CXCR2. The activation of ACKR1 leads to ERK MAPK activation [190], which inhibits CXCR2-dependent cell migration. Therefore, ACKR1 has the potential to be a therapeutic target for treating various cancers, particularly if both ACKR1 and CXCR2 are expressed in tumor cells.

The low expression of ACKR1 in breast cancer is strongly associated with intense tumor angiogenesis, ER status, lymph node metastasis, distant metastasis, and poor survival [186,191,192]. In a Chinese population, breast cancer patients with the FYa^+^FYb^+^ erythrocyte phenotype had a lower incidence, less frequent lymph node metastasis, and better prognosis than patients with the FYa^−^FYb^−^ phenotype [193]. Moreover, the rs12075 polymorphism in ACKR1 has been linked to worse relapse-free survival in patients with triple-negative breast cancer [194] and lymph node metastasis in patients with breast cancer [195]. These findings suggest that ACKR1 expression levels and genetic variations in ACKR1 may serve as important prognostic markers for breast cancer.

## 8. Anticancer Drugs Targeting CXCL1

CXCL1 plays a significant role in the development and progression of breast cancer, cervical cancer, endometrial cancer, ovarian cancer, and prostate cancer. For this reason, new anticancer drugs targeting this chemokine are being tested, for example HL2401, a monoclonal antibody anti-human CXCL1 [179]. HL2401 inhibits the autocrine growth of prostate cancer cells and also exhibits antitumor effects on animal models of this cancer. Inhibitors of the CXCR2 receptor are also being tested. CXCR2 is a receptor for CXCL1 and other CXC chemokines, including CXCL8/IL-8 [4]. The most extensively studied CXCR2 inhibitor is SB225002: (N-(2-hydroxy-4-nitrophenyl)-N′-(2-bromophenyl)urea) [181], which exhibits antitumor activity against cervical cancer [107], ovarian cancer [196], and androgen-independent prostate cancer [180]. In addition to inhibiting CXCR2, SB225002 also binds to β-tubulin, leading to the disruption of microtubule function [197,198].

Targeting the CXCL1-CXCR2 axis may be significant in other therapeutic approaches. Adoptive cell therapy is being tested, which involves taking lymphocytes from the patient, modifying them, and then reintroducing them into the patient’s body [199]. However, lymphocytes do not express high levels of CXCR2 [200]. Therefore, they do not migrate to sites where the levels of CXCR2 ligands are high. By increasing the expression of CXCR2, for example through transduction, such cells can specifically migrate to the tumor site with high levels of CXCL1 and other CXCR2 ligands [121]. As a result, they may be more effective in the fight against the tumor, as shown in experiments on ovarian tumors [200].

Targeting CXCR2 as a therapeutic approach can potentially augment the effectiveness of the existing standard anticancer therapy, owing to the ability of CXCL1 and other CXCR2 ligands to induce resistance to treatment. CXCL1 expression is upregulated by anticancer therapies, particularly by radiotherapy [97] and chemotherapeutics such as doxorubicin [99], oxaliplatin [182], and paclitaxel [98]. Consequently, CXCL1 activates its receptor CXCR2, triggering the activation of NF-κB [201]. This transcription factor directly increases the expression of anti-apoptotic Bcl-2 family proteins such as Bcl-2 [201], Bfl-1/A1 [202], and Bcl-xL [203], which inhibit apoptosis induced by chemotherapeutic agents. Additionally, CXCL1 recruits G-MDSC [52,57], which contribute to chemoresistance by secreting S100A8/9 [28]. Therefore, targeting CXCL1 and CXCR2 can enhance the effectiveness of the current standard anticancer therapy, particularly for doxorubicin [204], paclitaxel [204], and oxaliplatin [201].

Currently, only one clinical trial targeting CXCR2 is available in the PubMed database (https://pubmed.ncbi.nlm.nih.gov/, accessed on 31 March 2023) [205]. In this study, patients with HER2-negative breast cancer received reparixin, a CXCR1/CXCR2 antagonist, prior to surgical tumor removal. The experimental drug was safe and well-tolerated, and in some patients, reparixin reduced the number of breast cancer stem cells [205]. The ClinicalTrials.gov NIH U.S. National Library of Medicine website (https://clinicaltrials.gov/ct2/home, accessed on 31 March 2023) provides information on four ongoing clinical trials targeting CXCR2 (Table 6). Two of these trials (NCT03161431 and NCT04477343) are phase 1 studies evaluating the safety of SX-682, a CXCR1/CXCR2 inhibitor. Another clinical trial (NCT04552743) does not investigate the antitumor effect but the efficacy in mobilizing hematopoietic stem cells for future transplantation. The final clinical trial (NCT01740557) investigates CXCR2-transduced autologous T cells, particularly safety and clinical response in patients with melanoma.

## Figures and Tables

**Figure 1 ijms-24-07262-f001:**
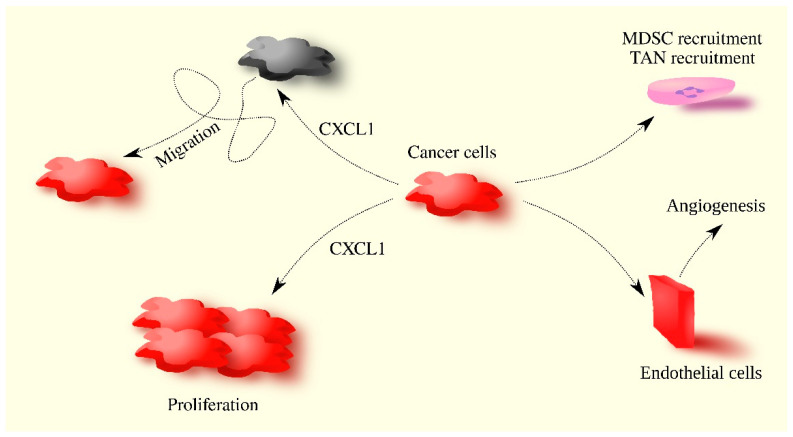
CXCL1’s involvement in cancer processes. CXCL1 increases proliferation and migration, and also acts on tumor-associated cells. CXCL1 recruits neutrophils, MDSCs, and induces angiogenesis.

**Figure 2 ijms-24-07262-f002:**
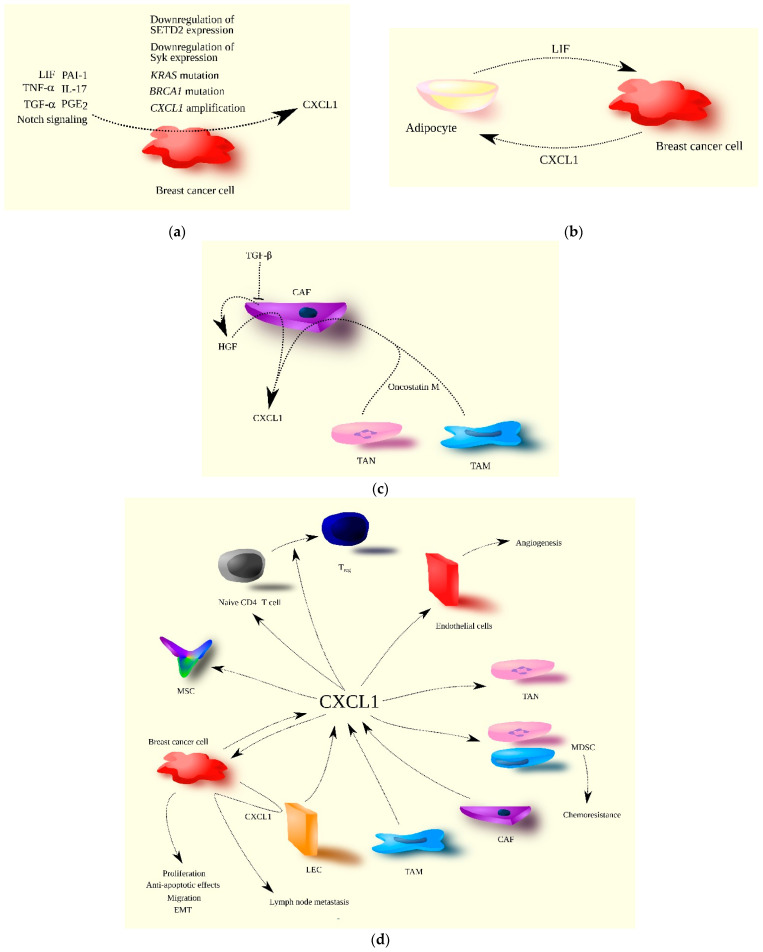
The significance of CXCL1 in breast cancer progression. (**a**) CXCL1 expression is increased in breast cancer cells by both extracellular and intracellular factors. TNF-α, TGF-α (EGFR ligand), PGE2, IL-17, LIF, PAI-1, and Notch signaling are examples of extracellular factors, while mutations in genes such as *KRAS* and *BRCA1*, *CXCL1* gene amplification, and the decreased expression of SETD2 and Syk are intracellular factors increasing CXCL1 expression in breast cancer cells. (**b**) The interaction of breast cancer cells with tumor-associated adipocytes results in increased CXCL1 expression in breast cancer cells due to the secretion of LIF by adipocytes. In turn, CXCL1 increases LIF expression in adipocytes, creating a positive feedback loop between these two types of cells in the breast tumor. (**c**) In the breast tumor, CXCL1 is secreted by CAF, and its expression in CAF is increased by oncostatin M produced by TAM and TAN. Increased CXCL1 expression in CAF is also due to their loss of responsiveness to TGF-β, resulting in increased HGF production by CAF, which further increases CXCL1 expression. (**d**) CXCL1 plays multiple roles in breast cancer progression, acting on various types of cells. Source of CXCL1 in breast tumor may be breast cancer cells, CAF, TAM, and LEC. CXCL1 promotes breast cancer cell proliferation, migration, EMT, and anti-apoptotic effects. Breast cancer cells also increase CXCL1 expression in LEC, promoting the migration of breast cancer cells to lymphatic vessels, resulting in lymph node metastasis. CXCL1 also acts on tumor-associated cells. It recruits immune cells such as naïve CD4^+^ T cells, TAN, MDSC, and MSC to the tumor niche. It can differentiate naïve CD4^+^ T cells into T_reg_ and facilitate chemoresistance through the recruitment of MDSC. CXCL1 also promotes angiogenesis via its influence on endothelial cells.

**Table 1 ijms-24-07262-t001:** Effect of CXCL1 expression level on survival of breast cancer patients.

Type of Cancer	Expression Testing Method	Impact on Survival at High CXCL1 Expression	Number of Patients in the Study	Notes	Source
Breast cancer: ERα-positive breast cancer	qRT-PCR	Worse prognosis	48	RFS	[29]
Breast cancer: ERα-positive breast cancer	Microarray Kaplan–Meier Plotter database	Better prognosis	2061	RFS Analysis based on the Kaplan–Meier Plotter database	[31]
Breast cancer	IHC	Worse prognosis	655	OS, nuclear CXCL1 expression, at cytoplasmic CXCL1 expression there was only a trend (*p* = 0.08)	[100]
Breast cancer	Microarray UALCAN/TCGA database	Better prognosis	1066	OS, Analysis based on UALCAN	[32]
Breast cancer	Microarray	Worse prognosis	121	OS	[34]
Breast cancer	Microarray Kaplan–Meier Plotter database	Better prognosis	3951/RFS 1402/OS	OS, RFS analysis based on the Kaplan–Meier Plotter database	[30]
Breast cancer	Microarray Kaplan–Meier Plotter database	No significant impact on prognosis	1402	OS, trend of worse prognosis at high CXCL1 (*p* = 0.064), based on the Kaplan–Meier Plotter database	[31]
Breast cancer	ELISA	Worse prognosis	61	OS, PFS circulating level of CXCL1	[38]
Breast cancer: basal breast cancer	Microarray Kaplan–Meier Plotter database	Worse prognosis	54	OS, based on the Kaplan–Meier Plotter database	[52]
Breast cancer	Microarray Finak microarray database	Worse prognosis	53	RFS stromal CXCL1 expression based on Finak microarray database	[37]

ELISA—enzyme-linked immunosorbent assay; IHC—immunohistochemistry; OS—overall survival; PFS—progression-free survival; RFS—relapse-free survival; qRT-PCR—quantitative real-time polymerase chain reaction. Red color—worse prognosis, Blue color—better prognosis.

**Table 2 ijms-24-07262-t002:** Effect of CXCL1 expression level on survival of cervical cancer patients.

Type of Cancer	Expression Testing Method	Impact on Survival at High CXCL1 Expression	Number of Patients in the Study	Notes	Source
Cervical cancer	Microarray GEPIA/TCGA database	Worse prognosis	292	OS based on the GEPIA database	[103,110]
Cervical cancer	IHC	Worse prognosis	150	DFS, trend of worse OS (*p* = 0.118).	[108]
Cervical cancer	IHC	Worse prognosis	40	OS	[104]

DFS—disease-free survival; IHC—immunohistochemistry; OS—overall survival. Red color—worse prognosis.

**Table 3 ijms-24-07262-t003:** Effect of CXCL1 expression level on survival of endometrial cancer patients.

Type of Cancer	Expression Testing Method	Impact on Survival at High CXCL1 Expression	Number of Patients in the Study	Notes	Source
Cervical cancer: uterine corpus endometrial carcinoma	Microarray GEPIA/TCGA database	No significant impact on prognosis	172	OS, DFS based on the GEPIA database	[110]

DFS—disease-free survival; OS—overall survival.

**Table 4 ijms-24-07262-t004:** Effect of CXCL1 expression level on survival of ovarian cancer patients.

Type of Cancer	Expression Testing Method	Impact on Survival at High CXCL1 Expression	Number of Patients in the Study	Notes	Source
Ovarian cancer	Microarray GEPIA/TCGA database	No significant impact on prognosis	212	OS, based on the GEPIA database	[110,121]
Ovarian cancer	ELISA	Worse prognosis	26	OS, serum level of CXCL1	[124]
Ovarian cancer	Microarray Kaplan–Meier Plotter database	Better prognosis	1656	OS, PFS based on the Kaplan–Meier Plotter database	[120]

ELISA—enzyme-linked immunosorbent assay; OS—overall survival; PFS—progression-free survival. Red color—worse prognosis, Blue color—better prognosis.

**Table 5 ijms-24-07262-t005:** Effect of CXCL1 expression level on survival of prostate cancer patients.

Type of Cancer	Expression Testing Method	Impact on Survival at High CXCL1 Expression	Number of Patients in the Study	Notes	Source
Prostate cancer	IHC	Worse prognosis	248	DFS, trend of worse OS	[177]
Prostate cancer	IHC	Worse prognosis	118	RFS	[154]
Prostate adenocarcinoma	Microarray GEPIA/TCGA database	Better prognosis	492	DFS, statistically insignificant difference in OS	[110]

DFS—disease-free survival; IHC—immunohistochemistry; OS—overall survival. Red color—worse prognosis, Blue color—better prognosis.

**Table 6 ijms-24-07262-t006:** Clinical trials available on the ClinicalTrials.gov NIH U.S. National Library of Medicine website.

Type of Cancer	Investigated Compound/Therapeutic Approach	ClinicalTrials.gov Identifier	Phase	The Aim of the Study	The Estimated Completion Date of the Clinical Trial
Metastatic melanoma stage III and IV	SX-682 (CXCR1/CXCR2 inhibitor)	NCT03161431	Phase 1	Safety analysis of SX-682 alone and in combination with pembrolizumab	December 2023
Multiple myeloma	MGTA-145 (CXCR2 agonist)	NCT04552743	Phase 2	Efficacy analysis of MGTA-145 with plerixafor in mobilizing hematopoietic stem cells	30 June 2022
Metastatic pancreatic ductal adenocarcinoma	SX-682 (CXCR1/CXCR2 inhibitor)	NCT04477343	Phase 1	Safety analysis of SX-682 in combination with nivolumab	31 December 2024
Metastatic melanoma, stage III and IV cutaneous melanoma	CXCR2-transduced autologous T cells	NCT01740557	Phase 1 and 2	Safety and clinical response analysis.	31 January 2024

## Data Availability

Not applicable.

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
