# Peer review of "Involvement in Tumorigenesis and Clinical Significance of CXCL1 in Reproductive Cancers: Breast Cancer, Cervical Cancer, Endometrial Cancer, Ovarian Cancer and Prostate Cancer"

_ijms, 2023, doi:10.3390/ijms24087262_

Round 1
Reviewer 1 Report
In the review manuscript, the authors mentioned the role of CXCL1 among various cancers. As a general review of the introduction of the topic, the content is sufficient, but some basic information should be added (see below). In addition, some of the information is too general, e.g. In line 414, “CXCL1 expression is elevated in endometrial tumours relative to healthy tissue [89]. Also, CXCL1 levels in the serum of patients with endometrial cancer are higher than in healthy subjects [83].” The details of the previous study were not mentioned. In line 268, “CXCL1 is also important in breast cancer metastasis.” More detailed information would be needed. Therefore, a revision will be needed. A major revision is recommended.
1. It will be good to have a diagram to illustrate the general molecular mechanisms driven by CXCL1 among different cancers in the introduction and explain briefly the importance of the mechanisms. In addition, authors might consider including the physiological function of CXCL1 so that the readers may recognise the importance of CXCL1 easier.
2. In line 132, the authors mentioned that CXCL1 amplification was common in breast cancer. The authors would need to address whether this event is associated with a particular clinical feature, e.g. subtype, BRCA1/2 mutation.
3. The contents should be elaborated in lines 137 to 140 because it would be a specific mechanism to mediate the high expression of CXCL1 in breast cancer.
4. In table 1, the authors should indicate how CXCL1 expression was examined in the previous studies. Since the results were controversial, an explanation or speculation will be needed.
5. Similarly, the detection method of CXCL1 in Tables 2,3,4 should be included.
6. The authors may consider separate in vitro and in vivo sections in each cancer.
7. In section 7, the authors might include clinical trials targeting CXCL1 or its receptor. This information will further emphasise the importance of CXCL1 in cancer therapy.
Author Response
Rev.1.
In the review manuscript, the authors mentioned the role of CXCL1 among various cancers. As a general review of the introduction of the topic, the content is sufficient, but some basic information should be added (see below). In addition, some of the information is too general, e.g. In line 414, “CXCL1 expression is elevated in endometrial tumours relative to healthy tissue [89]. Also, CXCL1 levels in the serum of patients with endometrial cancer are higher than in healthy subjects [83].” The details of the previous study were not mentioned. In line 268, “CXCL1 is also important in breast cancer metastasis.” More detailed information would be needed. Therefore, a revision will be needed. A major revision is recommended.
The sentences have been corrected and additional information has been added.
- It will be good to have a diagram to illustrate the general molecular mechanisms driven by CXCL1 among different cancers in the introduction and explain briefly the importance of the mechanisms. In addition, authors might consider including the physiological function of CXCL1 so that the readers may recognise the importance of CXCL1 easier.
A diagram and basic information about CXCL1 have been added.
- In line 132, the authors mentioned that CXCL1 amplification was common in breast cancer. The authors would need to address whether this event is associated with a particular clinical feature, e.g. subtype, BRCA1/2 mutation.
The analysis of the level of CXCL1 amplification in the analyzed tumors has been added.
- The contents should be elaborated in lines 137 to 140 because it would be a specific mechanism to mediate the high expression of CXCL1 in breast cancer.
The fragment has been changed. In the cited article, the authors investigated the precise impact of the BRCA1-GATA3 complex on FOXC1 expression. They also demonstrated that changes in GATA3 expression affect CXCL1 expression. However, they did not investigate the exact mechanism.
- In table 1, the authors should indicate how CXCL1 expression was examined in the previous studies. Since the results were controversial, an explanation or speculation will be needed. 5. Similarly, the detection method of CXCL1 in Tables 2,3,4 should be included.
The method of analyzing CXCL1 levels has been added to the tables.
- The authors may consider separate in vitro and in vivo sections in each cancer.
Each chapter of the paper has the following structure:
- Introduction, cancer incidence statistics, key general information about the specific cancer
- Expression levels in the tumor compared to healthy tissue
- Mechanisms of increased CXCL1 expression in the specific cancer
- Impact of CXCL1 on proliferation, cancer stem cells
- Impact of CXCL1 on migration, metastasis
- Impact of CXCL1 on tumor-associated cells, angiogenesis
- Clinical aspects, a table describing the association between CXCL1 expression levels and prognosis.
We believe that the current section structure is appropriate, and rearranging the information may make the article difficult to read.
- In section 7, the authors might include clinical trials targeting CXCL1 or its receptor. This information will further emphasise the importance of CXCL1 in cancer therapy.
A paragraph about the use of CXCR2 inhibitors in anticancer therapy has been added.
Reviewer 2 Report
The authors do a good job of providing a review of CXCL1 in the context of cancer, including breast, cervical, endometrial, ovarian, and prostrate cancer. Care was taken to review how CXCL1 and the signaling pathways it activated influences cancer progression and how levels of CXCL1 correlated to patient prognosis. The authors author also discuss drugs targeting CXCL1 signaling and potential uses in the treatment of cancer in the context of the current literature.
Minor concerns:
The authors missed an opportunity by not discussing atypical chemokine receptors, like DARC or ACKR1, that bind CXCL1 and how these receptors might impact CXCL1's role in cancer. One does have to limit the scope of one's topic, but an acknowledgement of atypical chemokine receptors or brief discussion might provide a greater perspective.
Author Response
Rev.2.
The authors do a good job of providing a review of CXCL1 in the context of cancer, including breast, cervical, endometrial, ovarian, and prostrate cancer. Care was taken to review how CXCL1 and the signaling pathways it activated influences cancer progression and how levels of CXCL1 correlated to patient prognosis. The authors author also discuss drugs targeting CXCL1 signaling and potential uses in the treatment of cancer in the context of the current literature.
Minor concerns:
The authors missed an opportunity by not discussing atypical chemokine receptors, like DARC or ACKR1, that bind CXCL1 and how these receptors might impact CXCL1's role in cancer. One does have to limit the scope of one's topic, but an acknowledgement of atypical chemokine receptors or brief discussion might provide a greater perspective.
A chapter about DARC/ACKR1 has been added
Reviewer 3 Report
The authors summarized recent findings regarding the clinical/biological significance of CXCL1 in breast, cervical, endometrial and prostate cancers. Overall, the manuscript is well described by appropriate references. Please consider following points.
1. Contribution of CXCL1 to chemoresistance is relatively poor. Recent findings have indicated functional role of CXCL1 in chemoresistance in human cancers.
2. (minor) The term "breast (or cervical, rndometrial, prostate) cancer tumor" may be corrected into "breast cancer" or "breast tumor".
3. (minor) Regarding the subtypes of breast cancers, Luminal HER2 (ER+HER2+) type is also the major subtype.
Author Response
Rev.3.
The authors summarized recent findings regarding the clinical/biological significance of CXCL1 in breast, cervical, endometrial and prostate cancers. Overall, the manuscript is well described by appropriate references. Please consider following points.
- Contribution of CXCL1 to chemoresistance is relatively poor. Recent findings have indicated functional role of CXCL1 in chemoresistance in human cancers.
A chapter about CXCL1-induced chemoresistance has been added
- (minor) The term "breast (or cervical, rndometrial, prostate) cancer tumor" may be corrected into "breast cancer" or "breast tumor".
It has been revised according to the reviewer's recommendations
- (minor) Regarding the subtypes of breast cancers, Luminal HER2 (ER+HER2+) type is also the major subtype.
An additional subtype has been added. The name of the HER2-positive breast cancer subtype has been changed
Round 2
Reviewer 1 Report
The authors have already addressed all my concerns.